# Which Factors Influence Running Gait in Children and Adolescents? A Narrative Review

**DOI:** 10.3390/ijerph20054621

**Published:** 2023-03-06

**Authors:** Anthony Sudlow, Paul Galantine, Fabrice Vercruyssen, Nicolas Peyrot, Jean-Jacques Raymond, Pascale Duché

**Affiliations:** 1Impact of Physical Activity on Health Research Unit, Faculty of Sport Sciences, University of Toulon, Campus La Garde, 83160 Toulon, France; 2Mouvement-Interactions-Performance, MIP, UR 4334, Faculty of Sport Sciences, Le Mans University, 72000 Le Mans, France; 3Unité de Médecine et de traumatologie du Sport, CHITS Hôpital Sainte Musse, 83100 Toulon, France

**Keywords:** biomechanics, running, growth, maturation, children, adolescents

## Abstract

In recent years, running has dramatically increased in children and adolescents, creating a need for a better understanding of running gait in this population; however, research on this topic is still limited. During childhood and adolescence multiple factors exist that likely influence and shape a child’s running mechanics and contribute to the high variability in running patterns. The aim of this narrative review was to gather together and assess the current evidence on the different factors that influence running gait throughout youth development. Factors were classified as organismic, environmental, or task-related. Age, body mass and composition, and leg length were the most researched factors, and all evidence was in favour of an impact on running gait. Sex, training, and footwear were also extensively researched; however, whereas the findings concerning footwear were all in support of an impact on running gait, those concerning sex and training were inconsistent. The remaining factors were moderately researched with the exception of strength, perceived exertion, and running history for which evidence was particularly limited. Nevertheless, all were in support of an impact on running gait. Running gait is multifactorial and many of the factors discussed are likely interdependent. Caution should therefore be taken when interpreting the effects of different factors in isolation.

## 1. Introduction

Over the last few decades participation in running as a sport has greatly increased in children and adolescents. Running distances in this population have also increased with an exponential rise in marathon and even ultramarathon participation in youth (under 19 years of age) [1]. Although exercise and physical activity is of great importance for a healthy development, there is a concern that increased running will result in an increase in running-related injuries [2]. In particular, the increase in running distance could present a health risk for growth and development as the bone–muscle–tendon complex is still immature [3]. Children’s increased participation in running calls for further investigation of their running gait as it is likely influenced by a multitude of different factors during childhood and adolescence. These factors may be intrinsic to individuals, such as genetics, growth, maturation, and sex, or of an extrinsic nature such as footwear or surfaces. An understanding of how these factors influence running gait is essential in order to know how children’s gait will change under a variety of conditions and to be prepared to make informed decisions that will benefit development and minimise injury risk.

Running gait can be defined as the movement patterns and mechanical strategies that individuals adopt in order to run. It is generally studied by measuring kinematic variables (joint or segment angles), kinetic variables (ground reaction forces, joint forces and moments), and spatiotemporal variables (step rate, step length, etc). Amongst healthy individuals there is a high variability in running patterns, and it is recognised that runners can be separated into biomechanically distinct groups [4]. This high variability is likely a result of different gait strategies; however, the explanation as to why individuals employ different strategies is complex and multifactorial. Running gait has been well studied in adults [5,6,7], and a recent review has even proposed a simple model enabling the categorisation into one of five different running styles by identification of certain key characteristics that differentiate between running patterns [8]. There is, however, a paucity of research examining running biomechanics in children and adolescents.

Differences in running gait between healthy and pathological children have been investigated [9,10,11], but few studies have explained the reasons for changes occurring in running gait in typically developing youth. To date, the existing studies seem to have focused on describing the mechanics behind the developmental process of running [12,13,14,15,16,17] as opposed to identifying the factors responsible for the changes in running patterns during growth and maturation. It is likely that the influence of these factors contributes to the shaping of an individual’s unique running style. Furthermore, knowledge of these factors and their effects could contribute to a better understanding of the high variability in running patterns and could also provide information on potential risk factors for running-related injuries. The aim of this narrative review was to examine the existing literature and critically assess the key information concerning the factors that influence running gait during childhood and adolescence. In order to group factors together in a consistent manner, the Grand Unified Theory of sports performance proposed by Glazier [18] was used. The proposed model stipulates that patterns of coordination and control, reflected here by running gait, are influenced by organismic, environmental, and task-related factors and their interactions. Organismic factors can be described as physical, physiological, morphological, or psychological; environmental factors are external to the movement system; and task-related factors are specific to the task being performed. The factors discussed hereafter have therefore been classified in this manner.

## 2. Organismic Factors

### 2.1. Genetics

An in-depth review of the genetics of sport and exercise is not the aim of the current analysis; however, a few areas must be addressed on this topic as many of the ensuing factors owe a certain proportion of their effects to an individual’s genetic makeup. Indeed, genes have been said to have an important part to play in the explanation of individual variation in the growth and development of children and adolescents [19]. Height, weight, body size, and even strength all have a certain amount of heritability [20] and are therefore genetically predetermined to a certain extent. Of course, many of these variables, such as body mass and strength, are also susceptible to behavioural influences including dietary intake, energy expenditure, and physical activity. Furthermore, the timing and rate of both growth and maturation are also subject to a strong genetic influence [20]. It is therefore important to bear in mind that certain factors influencing the development of running gait are indirectly controlled and determined by genetic expression. However, to date, no studies exist regarding the influence of genetics on youth running gait, probably due to the need for very large cohorts and both expensive and complex analyses.

### 2.2. Biological Age

Age has an indirect effect on running gait, particularly during childhood and adolescence. With advances in time there are multiple changes in a child’s characteristics that subsequently influence their running pattern. Indeed, numerous studies have reported differences in running gait variables at different chronological ages [15,21,22,23,24,25,26]. However, a more insightful analysis can be obtained by taking biological age into account. Biological age refers to both growth and maturity status at a given chronological age [20]. With increasing biological age changes in bone, tendon, and muscle growth may impact running gait in children and adolescents [27]. Indeed, these changes will affect factors, such as body mass, body composition, leg length, and strength, with subsequent effects on running gait. It is also important to bear in mind that the timing and rate of growth and maturation vary considerably among individuals, leading to differences in gait regardless of chronological age [12]. Therefore, it is more relevant to assess biological age when studying running gait in children and adolescents as most intrinsic factors that influence running gait are in fact dependent on biological age.

Advances in growth and maturity (i.e., biological age) also lead to an increase in sex differences due to the associated changes in anthropometrics and body composition. The timing of these differences between boys and girls are also subject to the timing and rate of growth and maturation and in general do not occur at similar chronological ages.

### 2.3. Sex

Throughout childhood and adolescence, multiple sex differences appear leading to differences in running gait between boys and girls. Indeed, from early to late childhood, sex differences in both step rate and step length are negligible as leg length between girls and boys remains similar [28,29]. The fact that peak height velocity (PHV) occurs earlier in girls is likely a contributing factor. However, from the onset of PHV in boys, their leg length surpasses that of girls [28] and contrasts in spatiotemporal parameters become increasingly apparent, especially during adolescence [30]. For more information concerning the influence of leg length on running gait see 2.5.

Prior to female puberty, boys and girls of similar chronological age are also similar in body mass, body composition, and strength [31]. However, from the onset of puberty, sex differences occur, primarily attributed to differences in the levels of circulating hormones and eventually leading to increased strength in adolescent boys compared with adolescent girls [32]. These sex differences in body composition affect running kinetics during sprinting, resulting in greater mass-specific force production in boys, not only due to greater gains in muscle mass compared with girls but also due to the higher gains in body fat in the latter [25,26]. These contrasts in mass-specific force production also result in spatiotemporal differences at maximal velocity, notably greater step length in boys [25,26]. At submaximal speeds, sex differences in relative force production and the influence on spatiotemporal parameters is not apparent, or at least does not seem to have been investigated, in children and adolescents.

Sex differences in kinematic variables during running have not been demonstrated in children but do exist between boys and girls during jumping tasks [33,34]. Furthermore, differences have been demonstrated during running between adult men and women [35,36,37]. It is likely that kinematic differences between sexes accompany the increasingly apparent differences in body size, composition, and strength previously reported [31,38,39]. However, from exactly what age or development phase running kinematics start to differ significantly between boy and girls is uncertain and could be joint specific. Other sex differences that could lead to differences in running gait include those concerning ankle joint range of motion (ROM). Indeed, a greater ROM about the ankle could lead to sex differences in foot strike pattern (FSP). Grimston et al. [40] reported that ankle ROM at rest was generally greater in girls compared with boys aged 9 to 20 years. However, a causal relationship between ankle joint ROM at rest and FSP during running has not been demonstrated. Moreover, to date, multiple studies have indicated no differences in FSP between sexes throughout youth [22,41,42].

Many of the sex differences mentioned previously can in fact be attributed to differences in size as opposed to sex differences per se after appropriate normalisation or scaling. It is therefore likely that most of these differences become apparent from puberty onwards. However, in adults certain differences in joint motion (ankle, pelvis, and torso) between men and women have been shown to persist even after normalisation, suggesting that some sex differences are not simply size related [35].

### 2.4. Body Mass and Composition

Body mass has been reported to affect running gait in children. At moderate speeds, heavier adolescents tend to have lower step rates and longer step lengths compared with lighter individuals [30]. At maximal speeds, greater mass is also associated with lower step rates; it is suggested that amongst other factors this could be due to additional mass leading to increased contact times [23,24]. The increases in contact times with additional mass have been observed to stabilize around and after PHV but do not naturally decrease with further maturation [23,24].

When considering body mass, the ratio of lean mass to fat mass should also be taken into account. Indeed, increases in body mass due to increases in muscle mass are likely to result in greater step lengths when sprinting as a result of greater force production [25,43]. On the contrary, increases in body mass due to greater levels of fat mass have been hypothesised to have a negative effect on mass-specific force production, leading to decreases in step length and even greater increases in contact time [24,25,43]. Increases in both muscle and fat mass occur as individuals advance in maturity, with a marked increase observed at the onset of puberty in healthy children. However, abnormal increases in fat mass in pre- and post-pubertal children can also arise in cases of obesity. Overweight prepubescent children have been reported to display significantly greater contact times and step lengths when running at low speeds compared with normal weight children [44]. Children with this condition also manifest larger contact areas and higher peak pressure at multiple foot regions compared with healthy children [44,45]. Furthermore, the changes in running patterns due to the bearing of excessive weight is thought to predispose these individuals to injury (overuse or musculoskeletal) and exercise-related pain [46,47]. Body mass clearly has an influence on running gait in children, but the effects differ with body composition. Multiple studies have investigated these effects at maximal velocity, but there appears to be a lack of studies concerning the effects observed at low to moderate speeds and in healthy and overweight youth.

### 2.5. Leg Length

One of the main anthropometric parameters affecting running gait during growth is leg length. In general, taller children display lower step rates and longer step lengths than those that are shorter [15,30]. Independent of running velocity, increases in leg length during growth affect spatiotemporal variables such as step rate and step length. Increases in leg length lead to a decrease in step rate and a corresponding increase in step length [15]. However, the effect of leg length on step rate is not always straightforward. When sprinting, step rate or cadence has been observed to increase in early childhood, decrease during the years preceding PHV, and finally stabilise around and post PHV despite continual increases in leg length throughout this entire period [23,25,43,48]. It has been reported that the multiple changes in sprinting cadence with increasing maturity are associated with the phenomenon of “adolescent awkwardness” [23,25,43]. This phenomenon has been described as a temporary disruption of motor coordination during periods of rapid growth and is said to occur primarily in males [20,49]. During this period, step rate during sprinting decreases and contact time increases, provoking a certain delay in maximal speed development [23,25]. Although many of the factors accounting for adolescent awkwardness are not apparent, recent research suggests that the rapid changes in body proportions (e.g., somatotype) during the adolescent growth spurt could disturb proprioceptive abilities leading to the changes observed in spatiotemporal variables [50]. Adolescent awkwardness may also influence running gait at submaximal speeds, but results only seem to have been reported in relation to sprinting.

In adults, voluntary increases in cadence reduced foot strike angle (FSA) and even led to changes from rearfoot strike (RFS) to mid or forefoot strike (MFS or FFS) [51]. In growing children, the natural decrease in step rate due to increasing leg length could therefore provoke an increase in FSA and a progressive shift from FFS to RFS. Indeed, this could partially explain the results observed in the study by Latorre Román et al. [22] in which the proportion of RFS observed in three to six year old children increased from 47% to 92% in adolescents aged 15 to 16 years. However, the fact that a large proportion of RFS already exists in young children remains unexplained and other factors contributing to changes in FSP are almost certainly involved.

### 2.6. Strength

It has been well established in typically developing children that increases in strength occur progressively alongside advances in growth and maturation [20]. Muscular strength has a significant impact on various aspects of running gait, but research on this particular topic appears to be scarce in children and adolescents. In particular, insufficient strength is likely to affect lower-limb kinematics and lead to side-to-side asymmetries during running with an increased risk of injury [2]. In injury-free children and adolescents, lower-limb kinematics have been demonstrated to be highly symmetrical between limbs, especially in the sagittal plane [52], but further investigation in children and adolescents with deficits in strength are needed. One study that could offer some insight is that of Wild et al. [53], in which reduced hamstring strength in 10 to 13-year-old girls was demonstrated to lead to greater knee abduction and internal rotation during a horizontal jumping (leaping) and landing exercise. However, although these results indicate that lower extremity strength influences biomechanics in younger children, it is unclear whether landing mechanics are transferable to those used during running. Future research directly assessing the effect of lower-limb strength on running gait in children and adolescents is therefore necessary in order to move forward on this topic.

### 2.7. Posture

Postural control is an essential part of sensorimotor function that uses the visual, vestibular, and proprioceptive systems to assess and regulate the location and position of the body in space [54]. In relation to running, Albertsen et al. [55] reported that healthy children (10–14 years) with lower sensorimotor control displayed higher variability in foot progression angles (foot rotation) during barefoot running. The authors concluded that the impairment in sensorimotor control was due to trunk muscle weakness. Subjects with lower sensorimotor control displayed increases in backward lean and thoracic and lumbar curvature (via rasterstereography). The article in question was in fact a short communication and the authors themselves highlighted the need for more studies on this topic. Indeed, very little research exists specifically addressing the influence of posture on running gait in typically developing children and adolescents. However, deficits in postural control due to pathologies such as developmental coordination disorder [9,10,56] or cerebral palsy [11,57,58] provide evidence that posture does have an effect on gait in both walking and running. In typically developing children, postural control has been reported to increase with chronological age [59], but declines in motor control are observed in early adolescence. Children have been reported to exhibit less postural stability in anterior–posterior directions during quiet standing [60], and girls in particular display a decrease in knee control during and after puberty [61,62]. It is therefore likely that changes in postural control during childhood and adolescence will be accompanied by changes in running gait. Furthermore, certain phases of vulnerability may exist during which deficits in postural control could increase the risk of running injury [50]. An example of such a phase is the period of rapid growth during puberty that is associated with the phenomenon of adolescent awkwardness mentioned previously (2.5.).

### 2.8. Perceived Exertion

Effort perception has been described as a combination of sensations such as fatigue, aches, and strains derived from cardiopulmonary and/or peripheral factors during exercise [63]. With increasing exercise intensity there is a strong relationship between perceived exertion (PE) and physiological variables such as heart rate and oxygen consumption as documented in youth [64]. The existence of this relationship questions whether there could also be a relationship between PE and running gait. In support of this hypothesis, a study by Robertson et al. [65] reported similar results between children’s self-rated values of PE and those determined by visual observation of leg, arm, torso, and head movement during treadmill exercise. In other words, the level of PE appeared to be related to running kinematics. However, these results should be interpreted with caution, as the experimental protocol consisted of exercise at relatively low speeds (4.8 to 7.3 km/h) and increasing gradients (2.5 to 12.5%). The relationship observed between PE and kinematics could have been driven by the increases in gradient that are known to be associated with increases in both exercise intensity and movement rather than an effect of PE on running gait.

A study by Kung et al. [66] investigating the differences in PE between children, adolescents, and adults at speeds around the walk to run transition could offer some further insight into the relationship between PE and running gait. According to the results, children (10 to 14 years) were less effective than adolescents (15 to 17 years) and adults at perceiving differences in exertion at speeds close to the walk to run transition and therefore had trouble deciding whether running or walking was more favourable. The authors concluded that changes in running gait are not solely driven by an attempt to lower energy cost but also by an attempt to reduce effort perception. However, according to their findings it appears that the ability to use PE to regulate gait is still developing in children up until the end of puberty. Despite a lack of evidence, running gait appears to be influenced by effort perception, which in turn is influenced by growth and maturation.

### 2.9. Injury and Pain

Sports-related injury during youth is common and has been studied extensively [2,27,67,68,69,70]. However, research investigating the effects of previous injury on running gait is lacking. According to one of the few studies, a prior history of bone stress injury in adolescent runners has been associated with greater peak hip flexion in both limbs during the stance phase and a trend towards greater knee flexion when compared with uninjured runners [71]. Moreover, the increased peak flexion angles of previously injured adolescents tend to occur later in the stance phase [71]. It is hypothesised that these differences in sagittal plane kinematics could be due to compensatory movements that arise following injury causing children or adolescents to modify their running gait.

It is possible that injury-related pain is the cause for the initial change in running gait, which may persist even after the pain has subsided. As mentioned previously, research in this area is lacking; however, other conditions specific to youth such as juvenile idiopathic arthritis (JIA) or calcaneal apophysitis (Sever’s disease) also appear to affect running gait via biomechanical adaptations associated with the avoidance of pain and therefore could provide some insight. Compared with healthy children and adolescents, those with JIA have been reported to display significant differences in walking gait. These differences include reduced step lengths, reduced hip and knee extension, lower plantar flexion and lower maximal ankle power at toe off and increased anterior pelvic tilt [72]. This particular gait is often referred to as crouch-like gait due to the hyperflexion of the hip and knee joints at initial contact as well as reduced plantar flexion at toe off [72]. Crouch gait could be linked to an avoidance of full knee extension that occurs at initial contact and terminal stance in order to reduce joint pain [73]. Furthermore, this compensatory movement could continue to occur in JIA individuals no longer experiencing sensations of pain due to an increased sensitivity to mechanical and thermal stimuli, referred to as the secondary consequences of JIA [73]. When running, it is possible that these alterations in gait might be further exaggerated; however, to date, no research seems to have examined this question. Children and adolescents with calcaneal apophysitis have been reported to have significantly higher step rates and significantly lower step lengths during running when compared with healthy children [74]. Additionally, self-reported pain has been significantly correlated with maximal rearfoot pressure [74]. Similar to JIA, this juvenile condition could affect running gait through the use of compensatory movements to avoid or minimise pain. Indeed, a higher step rate and shorter step length may be adopted in an attempt to decrease peak loads under the heel, as reported in adults [75], and therefore reduce pain when running.

Previous injury and sensations of pain when running appear to affect running gait during youth by causing compensatory movements. However, the sensation of pain may not have the same impact with increasing age as changes in the perception of pain have been reported to occur throughout childhood and adolescence, with older children often having a higher pain tolerance [76]. Furthermore, the research in this area is very limited and the retrospective design of most studies limits the ability to provide cause and effect conclusions. In the case of youth with prior injury or calcaneal apophysitis for example, the question of whether injury or pain leads to changes in running gait or whether in fact running gait that differs from the norm leads to injury or pain remains.

### 2.10. Running History, Lifestyle, and Training

In this section, running history will be assessed as all running experience accumulated outside of a structured training programme. Training, on the other hand, will be assessed as all activities specifically implemented to improve performance or reduce injury risk through changes in running gait.

Certain aspects of running gait such as FSA seem to be influenced by running history, with more experienced adolescent runners displaying decreased dorsiflexion and a greater tendency for MFS or FFS [42]. Examples can also be taken from the Hadza population of central Tanzania where men with more running experience tend to display an MFS, whereas women and children with less experience display a RFS. It would seem that as children increase in age, changes in running history or experience influence their running gait. However, it is likely that running experience is limited by an individual’s lifestyle. Indeed, the differences observed in the Hazda population are likely due to a difference in activity or lifestyle, as men are required to run when hunting, whereas women and children gather plant-based foods and run much less [77]. The culture and lifestyle of a country or population is therefore likely to have a specific effect on running history and in turn on running gait. Of particular concern is the current increase in sedentary behaviour in children and adolescents [78,79], which could negatively influence running history and subsequently affect running gait. However, this hypothesis necessitates further investigation and the answer to how running gait would be affected remains to be confirmed.

Training has been observed to be effective in inducing changes to multiple physiological characteristics in children and adolescents [80,81,82], yet research investigating the trainability of running gait is scarce. Undertaking training programmes during childhood and adolescence could have an effect on running gait but the evidence is unclear. Step rate, for example, appears to be higher in male and female adolescent runners with more years of training [30]. Improved running times without any increase in VO_2max_ have also been reported in children, indicating that training programmes may lead to a more economical style of running [80]. Sprint-specific training in youth, such as the use of sleds, has also been observed to provoke changes in gait such as increases in peak horizontal force, average step rate, and vertical displacement of the centre of mass [83]. However, in the case of resisted sprint training, no effects were observed in prepubescent boys, questioning whether there is in fact an age before which the trainability of running gait is not effective. In contrast with the above, a study examining functional exercise training in children aged 9 to 14 years observed little to no changes in running kinematics [84]. It was suggested that this could have been due to subjects already exhibiting optimal kinematics at baseline, insufficient training volume, or the lack of a crossover between the chosen exercises and running gait.

Additionally, the amount of time spent training in order for differences in gait to become apparent is not clear [30]. Indeed, Petray and Krahenbuhl [85] observed no significant changes in spatiotemporal variables such as stride rate and length in 10-year-old males after 12 weeks of training. It is possible that 12 weeks is not enough time to observe significant changes or that adaptations occur rapidly in novice runners but not in runners that are already trained. It could also be that the training had no effect due to the boys being prepubescent as mentioned earlier. Finally, it could simply be that training-induced changes in those particular variables (stride rate, stride length, and vertical displacement) do not occur as quickly as changes in other running gait variables. In any case, it is important to acknowledge that training studies often have specific protocols and therefore the results cannot necessarily be generalised.

Another way of measuring the effect of training on running gait is to quantify and compare training volume. Training volume can be defined as the distance run on a weekly, monthly, or yearly basis, often referred to as mileage. In adults, Clermont et al. [86] were able to distinguish between higher and lower mileage runners based on differences in running kinematics. This could be an extremely interesting way of assessing the effect of training volume on children’s and adolescents’ running gaits. However, to date, no research seems to have considered this approach. In summary, an effect of training on running gait in children and adolescents could exist but evidence is lacking. Furthermore, caution should be taken in distinguishing the effects due to training and those due to advances in growth and maturation.

## 3. Environmental Factors

### 3.1. Footwear

Running shoes have become increasingly common for both adults and children; however, the use of footwear has been reported to affect children’s running gait [87]. Indeed, although data in children and adolescents are scarce, multiple studies conducted over the last decade provide substantial evidence for the influence of footwear on running gait. In prepubescent children (six to nine years), larger FSAs (increased dorsiflexion) and higher average rates of RFS were observed when running in footwear with a large heel-to-toe drop compared with running barefoot [88]. Furthermore, running shoes also led to longer step lengths, lower cadence, and in some cases greater step widths [88]. Very similar results concerning the first point of contact during foot strike and the same spatiotemporal parameters were reported in adolescents (13 to 18 years) wearing running shoes with a large heel-to-toe drop, indicating comparable effects of footwear with increasing age [89]. However, despite the use of footwear leading to the same type of effects in both children and adolescents, the magnitude of the effect may not be the same, as it has been observed that younger children are less likely to use an RFS when running in shoes compared with older children [90]. Running in shoes has also been reported to lead to other kinematic differences compared with unshod running in youth, such as a less vertical trunk, more extended knees and hips on landing [42], an increase in foot inversion/eversion, and a decrease in foot rotation [41]. Another effect of footwear on running gait appears to be a decrease in kinematic variability, in particular for FSPs, possibly due to a dulling of exterior stimuli. Indeed, Lieberman et al. [42] reported that only 32% of shod adolescents used varied strike types versus 72% of barefoot individuals. Shoes have also been reported to indirectly influence impact forces during running by increasing the occurrence of RFS, potentially leading to overuse injuries [88]. However, according to data in adults it would seem that excessive dorsiflexion resulting in large FSAs plays a more important role in the increase in impact forces at ground contact than shoes per se [91]. This question has not yet been studied in children and adolescents, although understanding the effect of the interaction between shoe and FSA on impact forces is fundamental in preventing injury.

Differences in shoe type can affect the extent to which running gait is altered. Running footwear with a large heel-to-toe drop amplify dorsiflexion at foot strike and increase the probability of RFS patterns in children and adolescents [88,89]. Minimalistic shoes on the other hand, such as racing flats or track spikes, lead to a decrease in FSAs and a corresponding increase in MFS/FFS patterns compared with running shoes with high stack heights [88,89]. However, minimalistic shoes do significantly modify running gait compared with barefoot conditions and should not be considered the same as unshod running. It has been suggested that constantly using running shoes with a large heel-to-toe drop might affect the ability to use a FFS in youths that are still developing their running gait [89]. Indeed, as footwear with large heel-to-toe drops result in a higher probability of RFSs, children and adolescents may not develop the appropriate musculature and ligamentous strength required to FFS [89]. Although this may not be an issue for general physical activity, it could be problematic for youth participating in competitive sports, especially track athletics where an MFS/FFS combined with a short contact time is associated with a better performance [17].

Habituation, either to footwear or barefoot conditions, also has a notable impact on the development of FSPs in children and adolescents [90]. Indeed, habitually barefoot individuals experience a decrease in the rate of RFS between 6 and 18 years for barefoot running, barefoot sprinting, and shod sprinting but not for shod running. On the other hand, habitually shod individuals increase their rate of RFS between 6 and 18 years for shod running but remain more or less stable in the other conditions, with moderate to low rates of RFS, especially when barefoot [90]. Furthermore, independent of age, habitually barefoot individuals do not appear to change FSP when running with or without shoes. Conversely, habitually shod children retain moderate to high rates of RFS when using shoes but drastically reduce these rates and switch to MFS or FFS when barefoot, probably due to pain or discomfort as they are not accustomed to unshod running [90].

### 3.2. Running Surface and Incline

The surface on which a child or adolescent runs can also influence their running gait due to variations in certain characteristics such as stiffness or incline. It has been observed that youths running on harder surfaces for example, are less likely to use an RFS and will tend to plantarflex, whereas the opposite is true on softer surfaces [42,90]. Similar results have also been reported in adults, leading investigators to conclude that surface hardness has a significant influence on FSPs [92]. With this in mind, a factor contributing to the high rates of RFS during shod jogging in both habitually shod and barefoot children could be that cushioned trainers mimic a softer surface. Increases in surface stiffness have been reported to lead to decreases in leg stiffness in adults, most likely in order to maintain consistent support mechanics [93]. Although the mechanisms behind the adjustments in leg stiffness were not uncovered in this study, it is likely that it involves changes in leg compression through increases in flexion and in extension. There does not seem to be any research addressing this particular question in children; therefore, whether or not they adapt their leg stiffness in a similar fashion remains unclear.

Another issue that falls under the current section on running surfaces is the question of treadmill versus overground running and what this could imply for data from studies using treadmills and future research. Indeed, multiple studies have compared treadmill and overground running and have identified a number of differences in adults [94,95]; however, studies in children are scarce. One existing study in youth demonstrated that kinematics between treadmill and overground running were relatively similar, but large differences in kinetics were observed, including more anterior ground reaction forces during treadmill running [96]. Furthermore, despite most kinematic variables being similar, participants used an MFS/FFS on the treadmill as opposed to an RFS when running overground. Although many existing treadmill protocols implement a 1% slope in order to compensate for the absence of wind resistance, which could have small but significant biomechanical consequences, this was not specified. Finally, it was deemed that the use of a treadmill might increase instability in younger children with a less mature neuromuscular system [96]. Therefore, in an untrained paediatric population, treadmill running cannot be considered as a substitute for overground running when analysing running gait. Caution should be taken when interpreting the results of studies using treadmills to analyse youth running gait as certain outcomes may not be applicable to overground running.

Surface incline has also been confirmed to affect an individual’s running gait. Vernillo et al. [97] reported that uphill running increased step rate, decreased flight time and step length and led to the progressive adoption of an MFS/FFS pattern. Downhill running, on the contrary, was reported to decrease step rate, increase flight time and step length and lead to the progressive adoption of an RFS pattern [97]. However, once again these effects appear to have only been confirmed in adults. Bearing in mind that children already have higher step rates and lower step lengths compared with adults, whether or not they would adjust their running gait in a similar manner when confronted with the same ascent or descent is unclear and does not seem to have been studied.

## 4. Task-Related Factors

### 4.1. Running Modality or Type

Running modality influences running gait throughout youth and the effects appear to be mediated by fatigue. Indeed, high-intensity repeated sprint-type running is likely affected to a greater extent by peripheral fatigue, whereas moderate intensity continuous running is probably associated with greater central fatigue [98]. Multiple studies have reported that there is less exercise-induced peripheral fatigue and greater central fatigue in children and adolescents compared with adults [99,100,101,102]. Furthermore, for a similar amount of central fatigue, adolescents have been reported to experience more peripheral fatigue than children, indicating that growth and maturation may have a gradual effect on the development of neuromuscular fatigue [102,103]. The effects of modality on running gait are therefore likely to differ according to an individual’s neuromuscular fatigue profile and thus according to their maturation status.

Concerning intermittent type running, a study by Ratel et al. [104] comparing repeated treadmill sprints between children and adults reported lower peripheral fatigue for children along with smaller decreases in velocity, step rate, mean force output and both peak and mean power output. Furthermore, when the recovery time was increased, (15 s to 180 s) all variables remained unchanged for children, whereas for adults, step rate, mean force output and both peak and mean power output continued to decrease significantly across the sprints. It could be hypothesized that for adolescents, results would lie somewhere between those of children and adults; however, this has not been confirmed. Although this study gives some insight into the effects of intermittent/short distance running, further research is still needed. Indeed, not only were multiple biomechanical variables and certain age groups not included, but moreover, a recent study has revealed contrasting evidence indicating that no kinematic alterations were observed during or after two different intermittent running sessions in adults [105].

During continuous running, the literature states that children appear to fatigue at a faster rate than adults, possibly due to a combination of factors causing an increase in the energy cost of running [106]. Increases in exercise duration have been reported to increase central fatigue, as demonstrated in adults in both running and cycling [107,108]. However, since children and adolescents have been reported to develop greater levels of central fatigue for the same effort compared with adults, it is conceivable that during prolonged running children and adolescents would incur greater changes in running mechanics. Only two studies seem to have examined the fatigue-induced changes in running gait due to continuous running. The results indicated that in well-trained adolescents, a high-intensity run to exhaustion led to significant increases in contact time, peak vertical ground reaction force, centre of mass displacement during stance, leg compression, relative load under the medial longitudinal arch and mean foot contact area. Additionally, significant decreases in flight time, leg stiffness, mean plantar pressure, and plantar flexor fatigue resistance were observed [109,110]. The greater fatigability of the plantar flexors was observed to elicit a change in FSPs, as reported in adults [111]. The interpretation of these results is however somewhat limited as there was no comparison to untrained youth or other age groups, and many of the findings are likely only applicable to continuous running at a high intensity (95% VO_2max_ speed) that may in fact imply a considerable amount of peripheral fatigue as opposed to central fatigue.

Running modalities (intermittent vs. continuous) have distinct effects on running gait in youth: the effects are mediated by the origin of fatigue (peripheral vs. central) but also depend on the interaction between modality and intensity of exercise. Furthermore, maturation status needs to be taken into account. However, further evidence is currently lacking.

### 4.2. Running Velocity

Running gait parameters are influenced by changes in velocity in both children and adolescents. Indeed, spatiotemporal parameters such as step rate and step length have been observed to increase simultaneously with speed, whereas contact time has been noted to decrease and flight time to remain more or less constant [15]. Velocity also affects other kinetic and kinematic parameters of running gait such as the torque, power, work, and angle observed at the hip, knee, and ankle joints. However, the effect of velocity on these parameters largely depends on the specific joint, the plane of motion, and the phase of the running cycle and only seems to have been studied in adults [112,113].

Interestingly, the effects of speed on FSP and FSA remain unclear. Indeed, Lieberman et al. [42] failed to observe any effect of speed on FSA in adolescents. Furthermore, Williams et al. [17] did not observe any differences in time trial performance between children running with significantly different FSAs (dorsiflexed vs plantarflexed), indicating that speed did not affect their FSPs uniformly. An explanation for the apparent null effect of velocity on FSPs could be that the contrasts in inter-individual results confound the overall result, leading to a group outcome that is not truly representative of the different individual responses. Indeed, in a recent study by Forrester and Townend [114] in adults, clusters were created according to how participants’ FSP and FSA responded to increasing velocity and three distinct groups were observed. One group of subjects had large FSAs at low velocities and transitioned to smaller angles at higher velocities (≥5 m/s), a second group kept large positive FSAs (RFS) throughout, and a third group, on the contrary, had small negative FSAs (FFS) throughout [114]. However, when all the groups were pooled together, there was an overall tendency for a reduction in FSA at velocities above 5 m/s, although the results were less pronounced. The effect of velocity on FSP therefore appears to be more complex than previously thought. Whether or not the different FSP behaviours described above are also observed in children and adolescents is unknown. Despite these outcomes, it is interesting to note that a strong correlation has been observed between faster running (better performances) and a tendency for MFS/FFS in children and adolescents [17,42], especially when sprinting [115].

An up-to-date model illustrating the various factors that influence running gait and the current knowledge on their impact in children and adolescents is displayed below in Figure 1.

## 5. Conclusions

This review has identified a large number of organismic, environmental, and task-related factors demonstrating an influence on running gait during childhood and adolescence. Due to the narrative nature of this review, it is difficult to conclude whether or not certain factors have a greater impact than others. Nevertheless, age, body mass and composition, and leg length presented the greatest amount of evidence, all of which was in favour of an impact on running gait, and the effects of these factors appeared to be the largest. Considering these findings, the increase in growth and maturation during youth and the associated changes in body characteristics could potentially have the greatest impact on running gait compared with other factors. This is perhaps unsurprising given the substantial changes in anthropometrics and body composition that occur throughout childhood and adolescence. Sex, training, and footwear were also well researched; however, whereas the findings concerning footwear were all in support of an impact on running gait, those concerning sex and training were inconsistent and warrant further investigation. The remaining factors displayed a moderate amount of evidence, with the exception of strength, perceived exertion, and running history, for which evidence was particularly limited. Even so, all findings were in support of an impact on running gait. Finally, genetics had no direct evidence demonstrating an influence on running gait.

Although research is currently lacking in many domains, it is likely that all the factors included in this review do play a role in the development of an individual’s running gait, albeit to varying degrees. Furthermore, running gait is multifactorial, and it is likely that many of the factors discussed are interdependent. Therefore, caution should be taken when interpreting the effects of different factors in isolation. It is also likely that the effects of certain factors could provide valuable insight concerning risk factors for running-related injuries; however, this is outside the scope of the current review. Finally, as demonstrated throughout this review, many aspects of running gait are poorly researched in children and adolescents, rendering the evaluation of many factors and their effects unclear.

## Figures and Tables

**Figure 1 ijerph-20-04621-f001:**
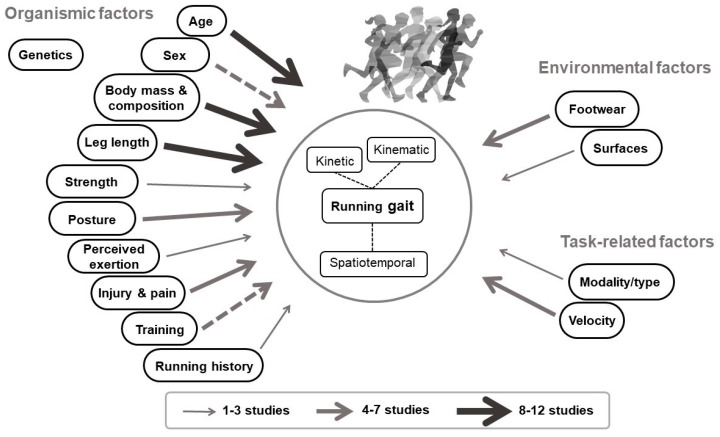
Factors influencing running gait in children and adolescents. Arrow size indicates number of studies. Full arrows indicate all studies are in favour of an impact on running gait. Dashed arrows indicate conflicting results. The running illustration is adapted from ©sabelskaya.

## Data Availability

Not applicable.

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
