# Peer review of "Which Factors Influence Running Gait in Children and Adolescents? A Narrative Review"

_ijerph, 2023, doi:10.3390/ijerph20054621_

Round 1

Reviewer 1 Report

General comment

The article proposes an interesting theme and highlights current knowledge as well as the unknown factors that influence the running gait in children and adolescents. The objective is ambitious because the variables that can influence the running pattern are numerous and it is complicated to give an overview of such a complex theme in a few pages, especially as the running gait varies enormously according to the type of effort made (sprinting versus endurance running). What is missing at the beginning of the article is a clear and synthetic presentation of what is meant by "running gait" and how running gait is generally studied in the literature.

For this narrative review, the authors propose to review the current state of knowledge by starting with factors defined as internal and external.  While this approach appears interesting and coherent, the choice of categories proposed is not clearly justified and sometimes raises questions because it does not allow the reader to fully understand the hierarchy of factors. For example, the genetic factor has a direct and more or less strong influence on various internal variables such as sex, body composition, body dimensions, but also neuromuscular and energetic characteristics. Age is also a factor that will influence body composition and size, neuromuscular and energetic characteristics. The personal experience of the individual (history, training but also injuries) is a factor that can influence the biomechanical (or technical), physiological (strength-speed, energy) and even psychological aspects. It appears that there are some more direct and some indirect factors that influence running gait and this should be highlighted in the review.

The extrinsic factors include elements related to the equipment (shoes), the environment (surface) and the task itself (running), which can be done in different modalities (speed, duration and continuous versus intermittent). This lack of consistency in the classification of factors makes the article difficult to read. The use of a global theoretical model, like the one proposed by Glazier (Glazier, P. S. (2017). Towards a grand unified theory of sports performance. Human movement science, 56, 139-156.) could be interesting to help the authors structure the presentation of the factors. The evolution of internal factors as a function of gender and biological growth rate is a crucial element in the evolution of running pattern and should be given special attention perhaps in an independent chapter.

Specific comments

Please add a list of abbreviations as there are many in the text.

References that follow each other should be grouped in the text, as recommended, and not separated by a comma.

L56-L58: reword the sentence as the current studies are mainly descriptive according age, but have not looked at the factors responsible for the evolution of biomechanical descriptors.

Genetics paragraph: explain why there are no genetic studies?

L76: "Environmental influences" is not adequate. Change to "behavioural"?

L110 : a reference is needed here.

L125 :  All growth curves show that changes between boys and girls stop to be parallel before 14YO !!! Author need here to be more accurate here and take into account the particularities of growth in girls and boys in this peripubertal period

L127-130 : it’s not clear why this hypothesis is reported here.

Posture paragraph: Shouldn't the chapter be about the development of coordination and running technique and include the notion of posture as a sub-element of coordination?

Perceived exertion: I’m not sur of the relevance of this chapter here because it is very specific to exhausting running tasks. Maybe you could develop this in a chapter focusing on specificities of running task and particularities of youth.

L386-388 : maybe explain here that treadmill running is mostly achieved with a 1% slope in order to compensate absence of  wind resistance. This difference may have small but significant biomechanical consequences.

L407-408 : a reference is needed here.

L467-469 : Results are surprising as small changes in velocities have individual significant consequence on running pattern.. Did they used paired statistics ?

L519-520 : results of this study are mainly showing that no significant changes at a standardized velocity is observed between three different treatments. This is a very specific context that cannot be generalised. Do you have information on the effectiveness of running technique training in 10 year olds children on maximum running speed, leg cycling, and consequences on biomechanical parameters ?

Conclusion should be adapted accordingly to the improvements made to the paper.

Reviewer 2 Report

Reviewing this article has been a pleasure. Apart from my interest in the topic, I consider it relevant in the current research connected to children and adolescents’ development. The comments below are provided in order to enhance the current manuscript.

Abstract

It is very well described.

Introduction

Did you consider including climatology characteristics as one of the extrinsic factors?

Conclusions

well described,..

Reviewer 3 Report

The document analyzes carefully and with a high level of precision the objective of the proposed study, also adequately describes each of the variables considered of interest in relation to the gait when running. Therefore, my final decision is to "accept" this manuscript and propose it for publication. Moreover, I would like to congratulate the authors for the quality of the work.
